# Potential Imaging Capability of Optical Coherence Tomography as Dental Optical Probe: A Mini-Review

**Ramadhan Hardani Putra** [1,2]**, Nobuhiro Yoda** [1,3,*]**, Eha Renwi Astuti** [2] **and Keiichi Sasaki** [1]

[1] Division of Advanced Prosthetic Dentistry, Tohoku University Graduate School of Dentistry, Sendai 980-0875, Japan; ramadhan.hardani@fkg.unair.ac.id (R.H.P.); keiichi.sasaki.e6@tohoku.ac.jp (K.S.)

[2] Department of Dentomaxillofacial Radiology, Faculty of Dental Medicine, Universitas Airlangga, Surabaya 60132, Indonesia; eha-r-a@fkg.unair.ac.id

[3] Department of Prosthodontics, Faculty of Dental Medicine, Universitas Airlangga, Surabaya 60132, Indonesia

[*] Correspondence: nobuhiro.yoda.e2@tohoku.ac.jp; Tel.: +81-22-717-8369; Fax: +81-22-717-8371

**Featured Application: Further development of OCT technology is expected to improve its feasibility for dental practice as a dental optical probe through high-resolution and non-invasive imaging without the use of ionizing radiation.**

**Abstract:** Optical coherence tomography (OCT) has been emerging in the dental field as an alternative diagnostic imaging for "optical probes" owing to its micro-meter resolution and non-invasiveness. This review aims to answer the following question: what is the imaging capability of OCT to visualize the subgingival area? Online searches were performed on PubMed and SPIE digital library databases, followed by a manual screening of references listed in relevant studies. The feasibility and imaging performance of OCT to visualize the subgingival area, including the periodontal, peri-implant, and crown margins, are discussed. All of the literature reviewed in this study demonstrated that OCT has the ability to visualize periodontal, including hard and soft tissues, and peri-implant conditions with high resolution. Gingival sulcus depth, periodontal pocket, and calculus deposition can also be depicted. However, clinical evidence that support the imaging capability of OCT as a dental optical probe to visualize subgingival area is lacking. Limited availability, portability, and usability of OCT for clinical experiments in dentistry, particularly for the subgingival area, might be contributed to its limitations. Hence, further development of handheld OCT systems and controlled clinical trials are needed to confirm the imaging capability of OCT reported in this review.

**Keywords:** diagnostic imaging; optical coherence tomography; subgingival

## 1. Introduction

Optical coherence tomography (OCT) is a non-invasive imaging tool that is emerging in various fields of medicine including dentistry. It can provide cross-sectional images of both hard and soft tissues with micrometric resolution [1]. Using the principle of low-coherence interferometry, OCT images are generated by measuring the intensity and time delay of the reflected or backscattered near-infrared light from the tissue structure, which is analogous to the underlying principle of medical ultrasound imaging. OCT can provide near real-time or video rate in situ images of tissues, owing to its high acquisition speed [2].

OCT can be used as an alternative to invasive diagnostic methods (e.g., biopsy or exploratory surgery), histological examination, and other imaging modalities. A comparison of OCT and other imaging methods is presented in Table 1. Although the principle of OCT is comparable to that of ultrasonography, both methods use different source images. OCT, which has limited tissue penetration, can provide a higher resolution than ultrasonography [3]. The avoidance of ionizing radiation can also be considered an advantage of OCT imaging. Owing to the aforementioned advantages, OCT has been used in many

medical and dental research projects aiming to optimize its utilization as an experimental and diagnostic tool.

**Table 1.** Comparison of optical coherence tomography (OCT) and other imaging modalities.

| Imaging Modality | Resolution (μm) | Penetration Depth | Source of Image |
|---|---|---|---|
| OCT | ~20 [3] | 1–3 mm | Near-infrared light |
| Medical Ultrasound | 500–1500 [2] | 10–20 cm | Ultrasound |
| Micro CT | ~50 [4] | Entire tissues | X-ray |
| CBCT | 80–600 [5] | Entire tissues | X-ray |
| Medical CT | 100–1000 [2] | Entire tissues | X-ray |
| MRI | 100–1000 [2] | Entire tissues | Magnetic field |

CT, computed tomography; CBCT, cone-beam CT; MRI, magnetic resonance imaging.

In dentistry, OCT has been widely used as an experimental tool rather than a diagnostic tool due to portability issues and its high initial cost. The use of OCT for clinical purposes in dentistry is also still limited. Therefore, several studies have attempted to optimize the advantages of OCT technology in dentistry. As a modern imaging tool, OCT can assist dentists to improve diagnosis by providing non-invasive high-resolution images in real time without the use of ionizing radiation. In conservative dentistry, OCT has been commonly applied to hard tissue or restoration material (that is, diagnosis of caries [6–9], detection of tooth cracks [8–10], and assessment of the tooth-restoration interface [8,11–13]). Compared to clinical and radiological assessments, OCT can improve caries detection due to its micrometer resolution images [14]. Moreover, OCT can distinguish tooth demineralization from healthy tooth tissues. It can also be used to detect the gap formation between the tooth and restoration material, which may lead to microleakage [8].

To date, OCT has been used to visualize periodontal tissues, including hard and soft tissues in the subgingival area [3,15]. Clinically, a subgingival condition is generally examined using a periodontal probe since radiographical assessment cannot accurately depict the structure of this area in sufficient detail. However, the probing method, which can be considered as an invasive method, can be painful and at times inaccurate since it is performed without visual guidance. Therefore, the purpose of this review was to discuss the potential of OCT as a dental optical probe and answer the focus question: "What is the imaging capability of OCT to visualize subgingival area?". In this review, the feasibility of OCT to visualize the subgingival area, including periodontal tissue, peri-implant condition, and marginal fit of prosthetic crown restorations have been discussed.

## 2. Search Strategy

Online searches were performed on PubMed and SPIE digital library databases. The combinations of search terms were constructed from the following: "optical coherence tomography", "OCT", "SD-OCT", "SS-OCT", "dental", "periodontal", "peri-implant", "gingiva" and "subgingival". This was followed by a manual screening of references listed in relevant studies. Due to explorative purposes and limited number of relevant reports, all of the studies that attempted to visualize the subgingival area using OCT regardless of the study design were reviewed. As a result, 12 studies investigated the visibility of periodontal tissue, peri-implant condition, and marginal evaluation of dental prosthesis were reviewed, including in vitro, in vivo, and ex vivo studies performed in animal and human subjects.

## 3. Application of OCT for Subgingival Area Visualization

### 3.1. Periodontal Tissues Condition

Table 2 summarizes the different studies that attempted to visualize and measure the morphometry of periodontal tissue using OCT. OCT has been successfully employed to generate images of periodontal hard and soft tissues at the subgingival area with microscopic detail [16–22]. A comparison of reported visualization of periodontal structures at the subgingival area from the reviewed literatures was shown in Figure 1. All of the

studies demonstrated that OCT is able to visualize enamel, dentin, dentino-enamel junction, free gingival margin, and epithelial and connective tissue of gingiva [16–22]. Visualization of attached gingiva [16,18–20,22], alveolar bone [16,19,20,22], and calculus deposition were also reported [16,17,19,22]. Park et al. compared ex vivo OCT images of beagle dogs' periodontal structure with micro-CT and histologic appearance. The overall shape of the tooth and surrounding soft tissues of images procured using OCT were similar to those of the histological appearance. In this study, OCT could not reveal the internal architecture of tooth and bone, which were clearly depicted in micro-CT images due to a much higher penetration range [17]. However, there are other reports demonstrating that it can successfully distinguish microstructural aspect of periodontal soft tissues in animal and human subjects. These studies documented that important periodontal parameters such as biological width [20], gingival thickness [16,22], gingival sulcus [16,17,19–21], and periodontal pocket can be measured using OCT [18,23]. It has also been reported that biofilm [22], plaque [16] and calculus deposition can be detected using this imaging technique [16,17,22,24].

**Table 2.** Imaging capability of optical coherence tomography (OCT) to visualize periodontal tissues based on available literature.

| No | Authors | Subjects | Subgingival Visualization | | | | | | | | | | Main Findings |
|----|---------|----------|----|----|-----|-----|----|----|----|----|----|----|----------------|
| | | | En | D | DEJ | CEJ | FG | AG | AB | Ep | CT | CD | |
| 1 | Mota et al. (2015) [16] | Porcines (ex vivo) | + | + | + | − | + | + | + | + | + | + | OCT can visualize periodontal structures. Longer wavelength shows a deeper tissue penetration. |
| 2 | Park et al. (2017) [17] | Beagle dogs (ex vivo) | + | + | + | − | + | − | − | + | + | + | OCT can generate high resolution cross-sectional images of superficial periodontal structures |
| 3 | Kim et al. (2017) [18] | Porcines (ex vivo) | + | + | + | − | + | + | - | + | + | − | OCT can visualize periodontal pockets as well as show attachment loss |
| 4 | Fernandes et al. (2017) [19] | Human (in vivo) | + | + | + | + | + | + | + | + | + | + | OCT potentially can evaluate periodontal tissues and measure gingival sulcus depth |
| 5 | Kakizaki et al. (2017) [20] | Human (in vivo) | + | + | + | − | + | + | + | + | + | − | OCT can visualize and analyze the morphological structure of periodontal tissues in details. |
| 6 | Lee et al. (2017) [21] | Human (in vivo) | + | + | + | − | + | − | − | + | + | − | OCT can be used to quantitatively measure gingival sulcus depth |
| 7 | Fernandes et al. (2017) [22] | Human (in vivo) | + | + | + | + | + | + | + | + | + | + | OCT can be used to identify periodontal structures in follow-up of PD treatments. |

En, enamel; D, dentin; DEJ, dentino-enamel junction; CEJ, cemento-enamel junction; FG, free gingiva; AG, attached gingiva; AB, alveolar bone; Ep, epithelial; CT, connective tissue; CD, calculus deposition; (+) visible, (−) not visible; PD, periodontal disease.

Since OCT has emerged as a promising approach toward determining periodontal microstructures (as reported through a number of studies), Fernandes et al. conducted a pilot study to investigate the potential of OCT as a tool to evaluate the treatment of periodontal diseases. In this study, periodontal conditions of 14 human subjects with periodontal disease were evaluated using OCT on days 0, 30, 60, and 90 following treatment. The results provided evidence that OCT was able to depict changes in gingival thickness, deposition of biofilm and calculus upon treatment, and change in depth of probing comparable to manual and automatic probing [22]. Due to aforementioned important findings, the implementation of OCT in clinical use may be expected as an alternative diagnostic tool

of conventional and automatic probe, the so-called "optical probe". Nevertheless, further controlled clinical trials are needed to provide stronger evidence.

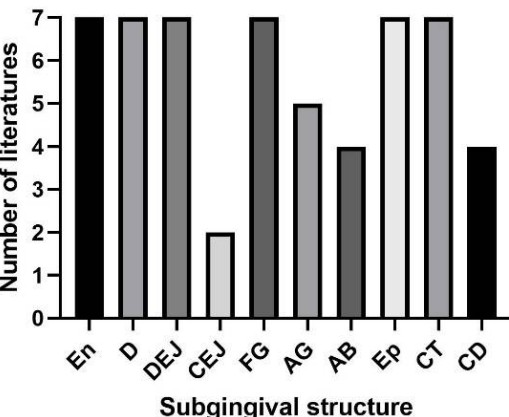

**Figure 1.** Comparison and distribution of visible subgingival structures reported from the included literatures. The score in the *y*-axis indicates the number of literatures that reported the visibility of the subgingival structures.

### 3.2. Peri-Implant Condition

The subgingival area is not only present around the natural tooth, but also in the peri-implant area. Radiographic examination and probing around the implant are considered as standard diagnostic tools for evaluating peri-implant conditions [25,26]. However, several shortcomings, such as metal artifacts [27] and underestimation of peri-implant bone loss [28], have been associated with radiographic examinations. Probing around implants is relatively technique-sensitive and can be affected by the presence of the abutment and prosthesis [29]. It can also cause more discomfort and pain compared to probing around the teeth [30,31]. Considering these limitations, the evaluation of peri-implant conditions through the use of OCT may provide essential additional information.

The potential use of OCT to evaluate peri-implant conditions has been demonstrated in animal subjects (ex vivo study). Sanda et al. reported that OCT images can clearly depict the implant body when the mucosal thickness is <1 mm. OCT images were also able to detect cement remnants at the submucosal area of the implant with a fixed crown when the sulcus depth was <2 mm and the mucosal thickness was <3 mm [32]. It has also been demonstrated that OCT images can be used to quantitatively measure peri-implant bone defects by comparing them with measurements using digital calipers as a reference (intraclass correlation coefficient = 0.99) [33]. Therefore, OCT may be considered as a potential novel imaging technique in implant dentistry, owing to its high resolution in peri-implant conditions. However, it should be noted that in vivo studies on human subjects have not been reported, and further clinical studies are required to provide evidence that OCT can be used to evaluate peri-implant conditions.

### 3.3. Evaluation of Dental Prothesis Marginal Adaptation

The crown margin, known as the subgingival margin, is often located in the subgingival area [34]. Since it is essential to ensure that the crown is well-adapted to the prepared abutment tooth, the marginal adaptation of the crown needs to be carefully evaluated. An inadequate marginal fit can lead to microleakage and plaque deposition, which in turn can induce secondary caries, gingival inflammation, or even periodontal disease. A marginal fit of ≤120 μm is acceptable in clinical practice [35], and this is within the resolution range of OCT. Hence, several studies have demonstrated the usefulness of OCT toward evaluating the marginal adaptation of interim/provisional [36,37] and ceramic crowns [38]. These in vitro studies used OCT images to assess the absolute marginal discrepancy [36,37] and

marginal gap [38]. However, and to the best of our knowledge, until now in vivo or ex vivo studies in either animal or human subjects have not been reported. Hence, the evidence of the applicability of OCT to evaluate marginal adaptation remains insufficient since it is not clear how periodontal tissues may influence the visibility of the subgingival margin area.

## 4. Limitation of OCT and Future Clinical Application

Some limitations should be considered when optimizing the capability of OCT in visualizing the subgingival area. Compensating for the ability to generate high-resolution, OCT has a limited tissue penetration depth (<3 mm), which is determined by the central wavelength and the numerical aperture of the collection optic [22,39]. Therefore, to obtain an optimum image, a different wavelength should be tested for imaging the targeted oral tissue, such as alveolar bone or gingival structures [3]. Additionally, a certain level of operator skills to align the device, capture the image, and ensure an interpretable image may also be critical [40]. Despite the promising results, a controlled clinical trial investigating the clinical applicability of OCT to image the subgingival area has not been reported until recently. Therefore, clinical evidence is still lacking. Limited availability and usability for clinical experiments, portability, and high cost of OCT are the likely factors contributing to this issue.

Nevertheless, the development of a handheld OCT system, especially for dental use, may improve the feasibility and reduce the initial cost in the future. A handheld OCT system can be beneficial for clinical trials and routine dental practice. Hence, it can attract many dental researchers to optimize its application. As a result, considering its potential, OCT may compliment other diagnostic imaging in dentistry and can improve diagnostic capability in daily practices. OCT technology may be applied to other imaging technologies, such as intraoral scanners [41] and dental radiography [42], resulting in a significant improvement in imaging performance. In particular, the integration of OCT technology into the existing oral scanner structure may enable optical impressions of abutment teeth that have been prepared in the subgingival. Such innovation will accelerate the current digitalization in dentistry. Finally, further developments in the dental field can transform OCT from an experimental tool to a standard diagnostic imaging modality for routine dental practices.

## 5. Conclusions

Although OCT can visualize periodontal tissue and peri-implant conditions with high resolution, there is no clear evidence that supports the imaging capability of OCT as a dental optical probe to visualize the subgingival area. This might be due to the limited availability, usability, and portability of OCT, coupled with lack of clinical trials in dentistry. Considering the advantages of OCT as discussed in this review, further development of handheld OCT systems, especially for dental use, is encouraged to improve its feasibility, reduce its expenses, and, more importantly, attract researchers to optimize its application in clinical practice. Hence, OCT is expected to improve diagnostic imaging performance in dentistry through high-resolution and non-invasive imaging without the use of ionizing radiation.

**Author Contributions:** Conceptualization, N.Y. and K.S.; data curation, R.H.P. and N.Y.; formal analysis, R.H.P. and E.R.A.; methodology, R.H.P., N.Y., and E.R.A.; funding acquisition, N.Y.; supervision, E.R.A. and K.S.; writing—original draft, R.H.P.; writing—review and editing, N.Y. and K.S. All authors have read and agreed to the published version of the manuscript.

**Funding:** This work was partially funded by Grant-in-Aid for Scientific Research (C: 19K10198) from Japan Society for the Promotion of Science.

**Institutional Review Board Statement:** Not applicable.

**Informed Consent Statement:** Not applicable.

**Data Availability Statement:** The data presented in this study are available on request from the corresponding author.

**Conflicts of Interest:** The authors declare no conflict of interest.

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
