# Peer review of "Potential Imaging Capability of Optical Coherence Tomography as Dental Optical Probe: A Mini-Review"

_applsci, doi:10.3390/app112211025_

Round 1

Reviewer 1 Report

The manuscript needs some corrections: 

line 16: Instead "How is the imaging..." please write "What is the imaging..."

line 47: medical and dental (not dan) 

line 73: write morphometry instead of morphometrical

line 93: too many as...please correct the sentence

line 103: write needed instead of warranted

line 135: write in italcs "in vitro" 

Reviewer 2 Report

Manuscript ID: applsci-1458954

Potential imaging capability of optical coherence tomography as dental optical probe: A mini-review

I think the topic of the topic is very good.

However, the importance of Optical coherence tomography (OCT) in dentistry should be discussed.

Secondly, even though it is A mini-review, it should be based on more reference articles.

The results of a simple table can hardly support the author's conclusion.
It should be said that it is difficult to get specific conclusions.

Round 2

Reviewer 2 Report

I suggest that you can use the References in Table 2 for analysis or comparison.
For example, summarize the distribution map and so on.
